# To Ubiquitinate or Not to Ubiquitinate: TRIM17 in Cell Life and Death

**DOI:** 10.3390/cells10051235

**Published:** 2021-05-18

**Authors:** Meenakshi Basu-Shrivastava, Alina Kozoriz, Solange Desagher, Iréna Lassot

**Affiliations:** Institut de Génétique Moléculaire de Montpellier, University Montpellier, CNRS, Montpellier, France; meenakshi.Basu@igmm.cnrs.fr (M.B.-S.); Alina.kozoriz@igmm.cnrs.fr (A.K.); solange.desagher@igmm.cnrs.fr (S.D.)

**Keywords:** TRIM17, ubiquitination, proteolysis, apoptosis, autophagy, mitosis, Parkinson’s disease, autism, cancer

## Abstract

TRIM17 is a member of the TRIM family, a large class of RING-containing E3 ubiquitin-ligases. It is expressed at low levels in adult tissues, except in testis and in some brain regions. However, it can be highly induced in stress conditions which makes it a putative stress sensor required for the triggering of key cellular responses. As most TRIM members, TRIM17 can act as an E3 ubiquitin-ligase and promote the degradation by the proteasome of substrates such as the antiapoptotic protein MCL1. Intriguingly, TRIM17 can also prevent the ubiquitination of other proteins and stabilize them, by binding to other TRIM proteins and inhibiting their E3 ubiquitin-ligase activity. This duality of action confers several pivotal roles to TRIM17 in crucial cellular processes such as apoptosis, autophagy or cell division, but also in pathological conditions as diverse as Parkinson’s disease or cancer. Here, in addition to recent data that endorse this duality, we review what is currently known from public databases and the literature about *TRIM17* gene regulation and expression, TRIM17 protein structure and interactions, as well as its involvement in cell physiology and human disorders.

## 1. Introduction

The tripartite motif (TRIM) family represents the largest subfamily of RING-containing E3 ubiquitin-ligases [1]. In humans, it includes more than 80 members [2,3,4]. TRIM proteins are characterized by the presence of a highly conserved tripartite motif at the N-terminus that is composed of a RING domain, one or two B-boxes (B1 and B2) and a coiled-coil (CC) domain. It is followed by a highly variable carboxy-terminal domain, which categorizes the different TRIMs into 12 distinct subgroups (C-I to C-XII) [1] (Figure 1). The subgroup C-IV is characterized by a PRY-SPRY domain that represents the most common C-terminal domain, accounting for about half of all known TRIMs [2,5,6]. The RING domain confers to TRIM proteins their E3 ligase activity [7], the other conserved domains being mostly involved in protein–protein interactions. The specific combination and order of the different domains within the tripartite motif, but also the spacing between each domain, is highly conserved. This suggests that the tripartite motif has been selectively maintained to perform specialized functions, such as ubiquitination, and represents a functional structure rather than a set of separate modules [1,8].

Ubiquitination involves the sequential action of an E1 ubiquitin-activating enzyme, an E2 ubiquitin-conjugating enzyme and an E3 ubiquitin protein ligase which specifically recognizes the substrate, binds an E2 enzyme and facilitates the covalent binding of ubiquitin (a 76-residue polypeptide highly conserved throughout evolution) mainly to a specific Lys residue of the substrate protein [9]. Ubiquitin itself can be ubiquitinated on different residues (K6, K11, K27, K29, K33, K48 and K63 or the N-terminal methionine residue) to form ubiquitin chains with many different conformations. Depending on the number of the substrate Lys residues that are modified, the number of ubiquitin molecules that are conjugated and the conformation of the chains formed, this cascade of reactions results in mono-, multi- or poly-ubiquitination that determine the fate of the substrates [10]. For example, K48-polyubiquitinated substrates are generally degraded by the proteasome [11], while K63-polyubiquitinated or monoubiquitinated substrates are rather eliminated by autophagy [12]. TRIM proteins, therefore, contribute to the efficient removal of short-lived or misfolded proteins and protein aggregates, mainly by the ubiquitin–proteasome system (UPS) but also by autophagy, and may participate in the crosstalk between these two systems [13].

TRIM17 (Tripartite motif-containing 17) also known as RNF16 (Ring Finger protein 16) or terf (testis RING finger protein) is part of the Class C-IV subfamily of TRIMs [1] (Figure 1). As such, TRIM17 contains a PRY-SPRY domain at its C-terminal extremity. TRIM17 was initially isolated from rat and human testis cDNA libraries in 1998 [14]. Both cDNAs encode a 477 amino acids protein that has been subsequently found to be expressed at low levels in various tissues [15]. This weak expression, which makes it difficult to detect with antibodies, may explain why TRIM17 was not studied for a long time. The E3 ubiquitin-ligase activity of TRIM17 was first demonstrated in 2009 [15] and its first cellular function was described in 2010, when we showed that mouse Trim17 is both necessary and sufficient for neuronal apoptosis [16]. Subsequently, we and others have identified different substrates and partners of TRIM17 and provided evidence that TRIM17 plays an important role in key cellular processes. Although it is low in most situations, TRIM17 expression can be dramatically induced following different cellular stresses, giving it the role of a sentinel ready to trigger the appropriate cellular response [15,17,18]. Here, we review the current knowledge on *TRIM17* gene regulation, substrates and partners of the TRIM17 protein, its cellular significance and the emerging role of TRIM17 in several human diseases.

**Figure 1 cells-10-01235-f001:**
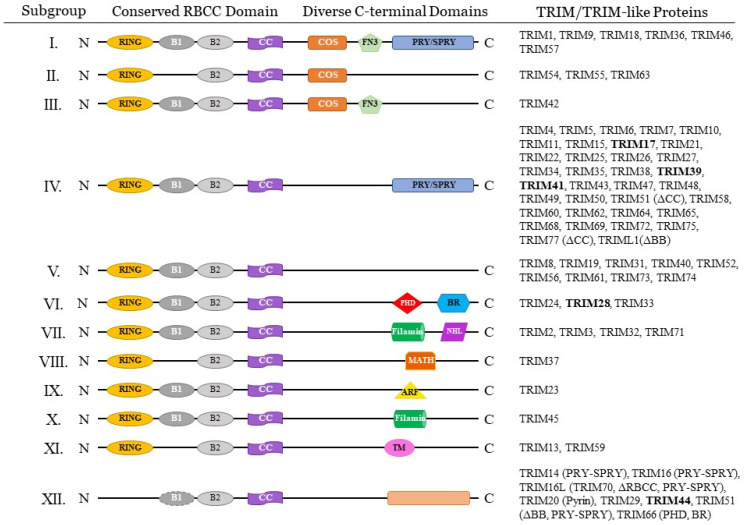
Classification of the TRIM proteins family. In bold, TRIM17 and its known TRIM partners described in this review. Adapted from [19,20].

## 2. Structure and Regulation of the *TRIM17* Gene

### 2.1. Structure of the TRIM17 Gene

Many *TRIM* genes exhibit the same genomic organization, with 6–7 exons spanning a 10 kb region [8]. The human *TRIM17* gene is located on 1q42.13 and spans less than 10 kb of the genome (University of California Santa Cruz (UCSC) Genome browser, genome build GRCh38/hg38, December 2013). In both humans and rodents, the gene is composed of 7 coding exons. In humans, its transcription seems to be driven by a single promoter and an alternatively spliced exon is present downstream the sixth coding exon (Figure 2). This could explain the existence of several TRIM17 transcript isoforms that involve not only the 3′ of the gene and several polyadenylation signals but also the coding region.

In both humans and rodents, *TRIM17* is located close to *TRIM11* (Figure 2). It is noteworthy that *TRIM11* also belongs to the subgroup C-IV of TRIM proteins and is the closest paralog gene of *TRIM17* [21] (data from ensembl.org, v.103 21 February 2021), with 46% amino acid sequence identity. These two genes probably derive from an ancestral gene duplication, *TRIM17* orthologues appearing 473 MYA ago in Gnathostomata and *TRIM11* orthologues about 158 MYA ago in Marsupials and Placental mammals [21] (data from ensembl.org, v.103 21 February 2021). However, *TRIM11* and *TRIM17* do not seem to share the same mechanisms of regulation. Indeed, according to data from GTEx Portal (GTEx Analysis Release V8, dbGaP Accession phs000424.v8.p2), *TRIM11* is more widely and highly expressed than *TRIM17*, although both have maximum expression in the cerebellum [17].

### 2.2. Transcription Factors and Regulatory Elements of the *TRIM17* Gene

The mechanisms and factors regulating *TRIM17* transcription are largely unknown. The transcription factor NANOG was shown to downregulate TRIM17 through upregulation of HDAC1 expression leading to HDAC1-mediated epigenetic repression of the *TRIM17* promoter. Therefore, TRIM17 expression seems to be regulated by HDAC1-mediated histone deacetylation [22].

We also found that NFATc3 and c-Jun transcription factors cooperate to induce *Trim17* transcription in neuronal cells [23]. Indeed, shRNA against NFATc3 and inhibition of c-Jun significantly reduced Trim17 mRNA levels in primary cerebellar granule neurons (CGNs) following apoptosis induction. In contrast, overexpression of NFATc3 induced Trim17 expression in mouse neuroblastoma Neuro2A cells. Furthermore, analysis of the *Trim17* promoter sequences revealed a region around the transcription start site containing two conserved AP-1 binding sites and one conserved composite NFAT:AP-1 element. We confirmed that c-Jun and NFATc3 indeed bind this regulatory region using chromatin immunoprecipitation (ChIP) assays. Interestingly, this binding strongly increased during early neuronal apoptosis [23].

In addition to these two studies, ChIP-seq data analysis from the ENCODE project (https://www.encodeproject.org, 21 February 2021) reveals the binding of several transcription factors, such as SUZ12, EZH2, AGO2 or RBM39, to regulatory elements in the *TRIM17* locus [24] (Bernstein and Snyder labs.). Indeed, chromatin immunoprecipitation using antibodies specific to these transcription factors, followed by sequencing of the precipitated DNA (ChIP-seq), indicate that they bind to the promoter region of *TRIM17*, around the transcription start site, delineated by the presence of CpG islands and specific histone modifications: H3K27Ac (acetylation of lysine 27 of the H3 histone protein), H3K4me1 (mono-methylation of lysine 4 of the H3 histone protein) and H3K4Me3 [24] (tri-methylation of lysine 4 of the H3 histone protein) (ENCODE 3 Nov 2018, Bernstein lab). These specific histone marks also suggest the presence of several cis-regulatory elements in the 5′ regions of the *TRIM17* gene, to which additional transcription factors can bind, such as STAT5 and ZNF592 [24] (Richard Myers, HAIB lab. and Michael Snyder, Stanford lab.). However, further studies are needed to determine whether these transcription factors and regulatory elements are indeed involved in *TRIM17* regulation and to elucidate the cellular context in which they operate. 

### 2.3. Induction of TRIM17 Expression and Regulatory Pathways: TRIM17 as A Stress Response Gene

As with almost half of the TRIM proteins, basal TRIM17 expression seems to be low. According to the GTEx Portal (GTEx Analysis Release V8), TRIM17 expression levels are generally less than 5 TPM (transcripts per million), except in a few tissues such as testis (Figure 3) [17,18,25]. Indeed, human TRIM17 cDNA was first isolated from a testis library and it was originally designated as testis RING finger protein (terf) because the corresponding transcripts were almost exclusively detected in human and rat testis using Northern blot analysis [14]. Apart from testis, TRIM17 has been found significantly expressed in spleen and thymus [15,17,18] and to a lesser extent in liver, and kidney [15]. Trim17 has also been reported to be ubiquitously expressed during mouse embryonic development (Reymond et al., 2001). Moreover, public databases and experimental data report a significant expression in some parts of the brain, in particular in the cerebellum (GTEx Analysis Release V8) (Figure 3) and in neurons of the substantia nigra [17]. However, it is possible that TRIM17 is ubiquitously expressed, at a very low level in most tissues, and that its expression is induced only in specific conditions.

Indeed, converging data indicate that *TRIM17* expression is induced by cellular stress. We initially identified *Trim17* as one of the most highly upregulated genes in primary cultures of mouse cerebellar granule neurons (CGNs) undergoing apoptosis following serum and KCl deprivation [16,25]. This well characterized in vitro model recapitulates the massive neuronal apoptosis that occurs during post-natal cerebellum development. Indeed, in the developing brain, immature neurons are produced in excess. Neurons that have not established the right connections are eliminated by apoptosis during the perinatal period to allow the formation of optimal neuronal networks [26]. CGNs require neurotrophic factors and stimulation from excitatory afferent neurons to survive in vivo. This can be mimicked in vitro by adding serum and a depolarizing concentration of KCl, the withdrawal of which triggers apoptosis. We showed that Trim17 is induced in serum and KCl-deprived CGNs, both at the mRNA and protein levels [16]. At the mRNA level, this induction can be enormous, reaching an increase factor of more than 50. Consistently, we observed that Trim17 protein is specifically expressed in apoptotic neurons, in the mouse cerebellum during postnatal development, its peak of expression coinciding with the peak of naturally occurring neuronal apoptosis [16]. Interestingly, Trim17 induction seems to be specific to transcription-dependent neuronal apoptosis induced by survival factor deprivation. Indeed, Trim17 levels are also increased in rat sympathetic neurons from the superior cervical ganglion (SCG) after NGF (neuronal growth factor) withdrawal and in motoneurons from mouse spinal cord dying in the absence of neurotrophic factors. In these two models, as well as in deprived CGNs, apoptosis can be prevented by the inhibition of transcription. However, Trim17 is not induced during CGN apoptosis triggered by low concentrations of glutamate, which is transcription-independent [16]. Induction of Trim17 following survival factor deprivation is triggered by two major signaling pathways associated with neuronal apoptosis [27,28]: activation of the proapoptotic kinase GSK3 due to the inhibition of the survival PI3K/Akt pathway, and activation of the JNK/c-Jun stress signaling pathway. Indeed, pharmacological inhibition of PI3K in survival conditions is enough to increase Trim17 mRNA levels, whereas inhibition of GSK3 or JNK prevents the increase in Trim17 expression triggered by serum and KCl deprivation in CGNs [16,23]. These data therefore suggest that *Trim17* is one the target genes of stress signaling pathways whose transcription is necessary to trigger neuronal apoptosis.

While survival factor deprivation mimics the physiological conditions encountered by neurons during brain development, other stresses such as exposure to drugs or environmental components can also induce abnormal TRIM17 expressions, which may favor human disorders. For example, Di-(2-ethylhexyl) phthalate (DEHP), which is widely found in plastics and is known for its reproductive toxicity and teratogenic effects, has been shown to activate TRIM17 protein expression through PPARγ (peroxisome proliferator-activated receptor γ) that plays a role in central nervous system (CNS) development [29]. This TRIM17 expression leads to caspase-3 activation and apoptosis [29]. This finding may account for the CNS toxicity of DEHP and implies that DEHP may impair fetal brain development. Consistent with stress-induced expression of TRIM17, the mRNA level of Trim17 is increased in the midbrain of mice treated with the neurotoxin MPTP (1-methyl-4-phenyl-1,2,3,6-tetrahydropyridine), a largely used animal model of Parkinson’s disease [30].

Viral or bacterial infections during development may also induce *TRIM17* in the brain. For example, *TRIM17* expression could be increased following inflammation-induced white matter injury (WMI) in preterm infants, which is associated with neurocognitive impairment and increased risk of neuropsychiatric disorders in adulthood. Indeed, in a robust model of WMI based on the injection of lipopolysaccharide (LPS) in the corpus callosum of 3-day-old (P3) rat pups, genomic DNA analyses revealed that postnatal inflammatory exposure causes hypomethylation of *Trim17* [31]. This epigenetic modification, which should result in increased Trim17 expression, is still present 3 weeks after the injury. Nevertheless, additional studies are required to determine the role of Trim17 in this disorder.

Taken together, current data indicate that TRIM17 is widely expressed at low levels but is induced in few tissues during physiological/developmental events and can be dramatically activated after drugs or environmental agent exposures. In many situations, this increase appears to be very strong and transient. TRIM17 may therefore serve as a sensor to trigger cellular stress responses. The identification of the transcription factors and regulatory elements that control its expression is all the more crucial.

## 3. TRIM17 Protein Structure and Molecular Function

### 3.1. TRIM17 Domain Composition and Species Conservation

TRIM proteins are metazoan-specific and have been identified in many species although their number is greatly increased in vertebrates [8]. As with other TRIM proteins, TRIM17 is characterized by the N-terminal tripartite motif consisting of a RING domain followed by one type 2 B-Box and a coiled-coil (CC) domain. TRIM17 has in its C-terminal part, a PRY-SPRY domain. The RING domain is a zinc-binding motif composed of 40–60 amino acids. It plays a critical role in the ubiquitination process by binding to E2 ubiquitin conjugating enzymes and by promoting the transfer of ubiquitin to the substrate [32] (Figure 4a). It can also mediate substrate conjugation with ubiquitin-like proteins such as SUMO [33] or NEDD8 [34]. Another key feature of the RING domain is its participation in homo- and hetero-dimerization [35,36]. The B-Box domains are zinc-finger domains that typically contain cysteine and histidine residues arranged in one of several motifs which are relatively conserved (Figure 4b). B-Box domains are a crucial feature of TRIM proteins, but they can be found in other protein families. The hyperhelical coiled-coil motifs can be found in numerous proteins where they have diverse functions [37]. The coiled-coil motif of TRIM proteins is around 100 residues long and is often broken up into two or three separate coiled-coil motifs (Figure 4c). A ‘rope-like structure’ stabilized by hydrophobic interactions is formed by alpha helices that are wound together in coiled-coil domains. Notably, the primary sequence of this region is not conserved [38]. The coiled-coil domain is mainly involved in homo- and hetero-interactions. It promotes the formation of high molecular weight complexes of TRIMs that define subcellular niches [39]. The PRY-SPRY domain represents the most common C-terminal domain, accounting for about half of all known TRIM proteins [2,5,6] (Figure 4d).

From an evolutionarily point of view, TRIM17 belongs to group II of the TRIM family, which is younger and faster evolving than group I [3] possibly because they are implicated in the innate immune response [3]. This group II is composed of 34 proteins (31 TRIM and 3 TRIM-like proteins), which possess only the B-box2 domain and are mostly organized as RING-B2-CC-SPRY proteins [3]. The TRIM genes of this group are present only in vertebrates, however, they are generally poorly conserved even within the same phylogenetic order where they play species-specific roles [3]. TRIM17 is present in most mammal species since 95 of the 108 placental mammal species have a TRIM17 orthologue. However, in contrast to other TRIM proteins belonging to the same subgroup such as TRIM39, TRIM17 is not present in reptiles, birds and fishes [21] (data from ensembl.org, 21 February 2021, release 103). Human TRIM17 shares 73%, 75% and 75% amino acid identities with its orthologues in rodents, cows and dogs, respectively [3], with variation rate depending on the domain. For example, the RING finger domain shares 84% identity while the B-box2 domain, the coiled-coil domain and the C-terminal domain share 78%, 60% and 80% identity, respectively, between human and rat TRIM17 [14]. These high homologies suggest the important role of these domains in TRIM17. Analysis of TRIM17 protein sequence conservation between different species highlights the amino acids that are essential for the structure and function of each domain, but also the regions that are important for TRIM17 specificity compared to other TRIM proteins (Figure 4, residues in green).

### 3.2. TRIM17 Isoforms

The full length TRIM17 protein comprises 477 amino acids both in humans and rat, with a molecular weight around 54.9 kDa in rat and 54.3 kDa in humans [14]. As in all TRIM genes, the tripartite motif of TRIM17 is encoded by a single exon, corroborating the evolution of this module as a single entity [8]. In humans, eight alternatively spliced transcript variants have been found for TRIM17 (Figure 2b), resulting in six different lengths of proteins [21] (data from ensembl.org, v.103 21 February 2021), which vary in their C-terminus (Figure 5). According to public databases, the full length TRIM17 protein is encoded by three different transcripts (Figure 5), whereas truncated proteins with 343, 267, 175, 50 and 30 amino acids are provided by five different transcripts [21] (data from ensembl.org, v.103 21 February 2021) (Figure 5). Of these isoforms, the existence of only the full length, 343 and 267 amino acids proteins has been demonstrated experimentally [40] (Figure 5a). 

The different TRIM17 transcripts seem to share a similar tissue expression profile with preferential expression in testis and brain (Figure 5b) (GTEx Analysis Release V8 (dbGaP Accession phs000424.v8.p2). According to data on GTEXportal, the human ENST00000456946 isoform, which codes for a 343 amino acids protein lacking its PRY-SPRY domain, represents the most expressed transcript in cerebellum, accounting for almost 30–40% of all TRIM17 transcripts, whereas in testis it accounts for less than 25% (GTEx Analysis Release V8 (dbGaP Accession phs000424.v8.p2, Figure 5b). However, the effective overrepresentation of this isoform, compared to the full-length protein, remains to be experimentally confirmed. The same TRIM17 isoform, lacking the PRY-SPRY domain, is also expressed in mouse.

The transcripts of other members of the TRIM family are also frequently alternatively spliced, producing different protein isoforms that often diverge at their C-terminal end [39]. Some of these TRIM isoforms exhibit different biochemical properties and activities. This has been well documented for TRIM19 and TRIM5alpha [41,42]. Furthermore, the concomitant, potentially tissue-specific presence of various isoforms of the same TRIM gene may, in some cases, involve cross-regulatory mechanisms. For example, the tripartite motif of TRIM28 has been shown to interact with the full-length protein, thereby generating a complex that is unable to bind its partner KRAB [43]. In the case of TRIM17, the possible effect of different isoforms has never been addressed and can only be speculated. 

### 3.3. Secondary and Higher Order Structures of TRIM17

While the C-terminal domain generally mediates target recognition and specificity of TRIM proteins [7,44] and the RING domain confers E3 ubiquitin-ligase activity, the B-box and especially the coiled-coil domains are involved in the formation of homo- or hetero-dimers or multimers [44,45,46,47]. 

#### 3.3.1. Monomer 3D Structure Model

Although over 400 RING motifs have been identified in the human genome and over 200 different three-dimensional (3D) structures are available in protein databases, relatively few structures have been resolved for TRIM proteins. These involve mainly NMR derived structures for certain domains of C-IV TRIMs. This is the case for B-Box, RING and PRY-SPRY of TRIM5alpha, the B-Boxes of TRIM21, TRIM39 and TRIM41 and the RING domains of TRIM34 and TRIM39 (PDB ID: 2YRG, 2ECV, 2LM3, 5JPX, 2DIF, 2EGM, 2EGP and 2ECJ, respectively) [48,49]. The 3D structure of TRIM17 has not been determined yet. However, the predicted monomeric 3D structure of human TRIM17 (UniProtKB ID: Q9Y577) can be generated in silico with SWISSModel (swissmodel.expasy.org, 25 February 2021) based on known structures and on the sequence homologies with the tripartite motif of TRIM28 (PDB ID: 6QAJ) [50] and the PRY-SPRY domain of TRIM20/PYRIN (PDB ID: 2WL1) [51] with which TRIM17 shares 18% and 35% sequence identities, respectively (Figure 6a). 

#### 3.3.2. Homodimerization of TRIM17

The ability of TRIM proteins to homo-interact through their coiled-coil region is one of their main structural features [39]. Using interaction mating technique and in vitro/and or in vivo coimmunoprecipitation techniques, Reymond et al. confirmed homo-interactions of TRIM1, 3, 5, 6, 8, 9, 10, 11, 18, 21, 23, 24, 25, 26, 27, 29, 30, 31, and 32. Although, we have shown that TRIM17 can also homo-interact [30], the domains that are involved in this process are still unknown and it is not clear whether this homo-interaction is required for its substrate binding or E3 ubiquitin-ligase activity. Several studies have reported that the tripartite motifs of TRIM5, TRIM25, TRIM28 and TRIM69 form antiparallel dimers [46,47,49,52,53]. If this structure is shared by all TRIM proteins, the predicted dimerization model for TRIM17 can be established from existing 3D homodimer structures of other TRIM proteins such as TRIM20/PYRIN and TRIM28 (from Swiss Model browser) (Figure 6b).

#### 3.3.3. Multimerization

Interestingly, for several TRIM members, higher molecular weight homocomplexes, compatible with the presence of more than two TRIM molecules, have also been observed [50,52,53,54,55,56]. These oligomers require the tripartite motif, the antiparallel structure of dimeric coiled-coil domains locating RING domains at opposite ends of the molecule and leading to RING interaction with other dimers to form higher order TRIM complexes [57]. Importantly, homo-multimerization has been suggested to be crucial for the function of several TRIM proteins, such as TRIM5alpha, PML or TRIM28 [50,54,55]. Notably, several studies have shown that homo-oligomerization of TRIM5alpha, TRIM25 or TRIM32 increases their E3 ubiquitin ligase activity [45,58,59]. This property to form dimers, tetramers and higher order structures could be conserved across the TRIM family. However, to date, no study has examined whether TRIM17 forms structures larger than dimers, such as trimers or hexamers, whether these structures are required for its functions and how this may impact its E3 ubiquitin ligase activity.

### 3.4. Hetero-Interactions of TRIM17 with Other TRIM Proteins

Many TRIM proteins have been shown to interact with other members of the family. For example, TRIM21 interacts with TRIM5alpha, leading to the ubiquitination and degradation of the latter in HEK293 cells [60]. This E3–substrate relationship between two TRIM proteins might be involved in other TRIM hetero-interactions either reported or postulated, such as those involving TRIM5alpha and a group of TRIM proteins (TRIM4, 6, 22, 27, and 34) that interestingly show colocalization within cytoplasmic bodies [61].

Many studies have reported interactions between TRIM17 and other TRIM proteins, especially with those containing a PRY-SPRY domain: TRIM39 [62,63,64,65,66], TRIM41 [30,64,67], TRIM5alpha and TRIM22 [68]; but also with members of other TRIM subgroups such as TRIM28 [69] and the RING-less protein TRIM44 [15,70] (Figure 7). Although many of these interactions have been identified in high throughput screens, a few have resulted from low-throughput focused studies that sought to elucidate their functional role. For example, TRIM44 has been found to bind TRIM17 in a yeast two-hybrid assay (Y2H) using the tripartite motif of TRIM17 as a bait to screen a prostate/breast cancer cDNA library [15]. In this study, TRIM44 was also shown to reduce the poly-ubiquitination of TRIM17, and to prevent its proteasomal degradation. This effect was probably mediated by the N-terminal region of TRIM44 that contains a zinc-finger domain found in ubiquitin hydrolases (ZF UBP) and ubiquitin specific proteases (USPs) [15]. In another Y2H screen, we identified TRIM41 as a putative partner of TRIM17, and the ability of the two proteins to interact with each other was confirmed by coimmunoprecipitation and proximity ligation assay (PLA) [30]. We have also characterized the interactions between TRIM17 and TRIM28 [69] and TRIM39 [65] by PLA and coimmunoprecipitation of both ectopically expressed and endogenous proteins. Importantly, in these three studies, TRIM17 appeared to inhibit the E3 ubiquitin ligase activities of TRIM28, TRIM39 and TRIM41 and the proteasomal degradation of their respective substrates [30,65,69]. Further investigations are needed to determine whether this inhibitory effect is a specific feature of TRIM17 and whether interaction of TRIM17 with other TRIM partners, such as TRIM5alpha and TRIM22, results in a similar outcome.

### 3.5. Ubiquitin Ligase Activity of TRIM17 and Interactions with E2 Enzymes

In ubiquitination, the specificity of the reaction is provided by the E3 enzymes that recognize the substrates, while the E2/E3 combinations determine the topology and length of ubiquitin chains [71]. While the genome has about 600 E3 coding genes, 35–40 genes encode putative E2 proteins. A direct interaction between E2 and E3 enzymes is required for the ubiquitin ligase reaction. We and others have demonstrated the E3 ubiquitin-ligase activity of TRIM7 by showing that both human and mouse TRIM17 can auto-ubiquitinate in vitro in the presence of specific E2 enzymes: UBE2E1, UBE2D2 and UBE2D3 [15,16]. In contrast to human TRIM17, which preferentially cooperates with UBE2E1 to induce poly-ubiquitination [15], mouse Trim17 also acts efficiently with the Ube2d2 and Ube2d3 and generates mostly mono-ubiquitination with Ube2e1. We identified Mcl1 as a substrate of Trim17 by showing that Trim17 can ubiquitinate recombinant Mcl1 in vitro in the presence of Ube2d2 and that Trim17 is involved in the ubiquitination of Mcl1 in neurons [72]. Then, TRIM17 has also been shown to induce the degradation of the kinetochore protein ZWINT by the UPS [73].

Different studies aiming at identifying E2/RING interactions [74,75] and more specifically E2/TRIM combination revealed a general preference of TRIM proteins for the D and E classes of E2 enzymes [76] (UBE2D and UBE2E, respectively). As a rule, TRIM17 has been shown to interact with more than one E2 enzyme, each of which can bind several other E3 ubiquitin ligases [64,74,76] (Figure 7). As mentioned above, TRIM17 can interact with several other TRIM family members. However, not all of these partners share exactly the same preference for E2 enzymes (Figure 7). Interestingly, while UBE2D2 appears to be a common partner for all TRIM17-binding TRIM proteins (with the exception of TRIM44 which lacks a RING domain), some E2 enzymes may be specific to certain dimers. Since binding to the correct E2 enzyme is a prerequisite for their E3 ubiquitin ligase activity, these differences in the E2 interaction profiles of TRIM17 partners could represent a supplementary level of regulation.

It will be very interesting in future studies to identify new substrates of TRIM17 and, for each of them, to describe their E2 specificities and ubiquitin chain topology. This may indeed reveal that TRIM17 exerts pleiotropic effects in cells. Furthermore, although not yet well understood, the ability of TRIM17 to form multimers suggests that the concomitant presence of different isoforms such as shorter forms lacking C-terminal domain, within the same cell may have important functional consequences.

### 3.6. Inhibition of Other TRIMs by TRIM17

In searching for new substrates of TRIM17, we unexpectedly found that TRIM17 inhibited rather than mediated the ubiquitination of certain proteins identified as its binding partners. It appeared that this effect was due to the inhibition of another E3 ubiquitin-ligase of the TRIM family. Indeed, we have shown in three independent studies that TRIM17 inhibits the ubiquitination and degradation (i) of the transcription factor ZSCAN21 mediated by TRIM41 [30], (ii) of the antiapoptotic protein BCL2A1 mediated by TRIM28 [69] and (iii) of the transcription factor NFATc3 mediated by TRIM39 [65]. Therefore, our current knowledge of the molecular function of TRIM17 suggests that it acts as much by inhibiting ubiquitination as by promoting it. The inhibitory effect of TRIM17 could result from several mechanisms that are not mutually exclusive.

First, TRIM17 could directly inhibit other TRIM proteins by forming inactive hetero-oligomers at the expense of homo-oligomerization which has been suggested to be necessary for the E3 ubiquitin ligase activity of TRIM proteins [45,58,59]. Indeed, TRIM17 physically interacts with TRIM41, TRIM28 and TRIM39 [30,65,69]. Formation of these hetero-dimers or hetero-multimers with TRIM17 may inhibit the intrinsic E3 ubiquitin-ligase activity of its TRIM partner, possibly by preventing the binding of the E2 enzyme. Indeed, we observed that TRIM17 prevents the auto ubiquitination of purified recombinant TRIM39 and TRIM41 in vitro [30,65]. Second, TRIM17 could prevent the binding of its TRIM partners to their respective substrates. Indeed, we have shown that TRIM17 reduces the interaction between TRIM41, TRIM28, TRIM39 and their respective substrates, as assessed by both coimmunoprecipitation of ectopically expressed proteins and PLA with endogenous proteins [30,65,69]. In the three cases, TRIM17 appeared to bind both the TRIM E3 ubiquitin-ligase and its substrate. Therefore, TRIM17 could reduce the TRIM/substrate interaction either by directly binding to the substrate in a competitive manner, or because hetero-dimer formation impedes the accessibility of the substrate to the TRIM partner. Further experiments are required to determine the structural determinants of the inhibitory effect of TRIM17 on other TRIM proteins. Nevertheless, it is unlikely that TRIM17 inhibits the ubiquitination of the substrates of its TRIM partners by associating with a deubiquitinating enzyme (DUB), as shown for other TRIM proteins [77,78]. Indeed, TRIM17 is able to inhibit the in vitro auto-ubiquitination of TRIM41 and the ubiquitination of NFATc3 mediated by TRIM39, in a completely acellular medium in the absence of any DUB [30,65]. It is also clear that TRIM17 does not inhibit its TRIM partners by inducing their ubiquitination and subsequent degradation because TRIM17 rather decreases the ubiquitination levels of TRIM39 and TRIM41 both in vitro and in cells [30,65]. Moreover, TRIM39 reciprocally decreases the in vitro auto-ubiquitination of TRIM17, further suggesting that TRIM17 and TRIM39 form inactive hetero-dimers or hetero-multimers, in which the E3 ubiquitin-ligase activity of the two partners is inhibited.

Taken together, existing data about the molecular function of TRIM17 clearly indicate that it is an E3 ubiquitin-ligase that is able to promote its own ubiquitination as well as the ubiquitination of specific substrates. However, TRIM17 is also able to inhibit the ubiquitination of specific proteins mediated by other TRIM protein, by binding both the substrate and its E3 ubiquitin-ligase. This intriguing feature is certainly related to the propensity of TRIM proteins to hetero-interact. A similar effect was reported for TRIM24 which inhibits the degradation of dysbindin induced by TRIM32 in cardiomyocytes [79]. Further studies will determine whether this kind of function is a general characteristic of TRIM proteins and whether TRIM17 can stabilize other substrates by acting on additional TRIM proteins. However, this duality, whereby TRIM17 can promote or inhibit the ubiquitination and degradation of specific proteins depending on the cellular context, already appears to be crucial for its cellular functions, as discussed below. 

## 4. Cellular Functions of TRIM17

Although only a few substrates and partners have been identified for TRIM17 so far, the impact of TRIM17 on these proteins places it as an important regulator in key cellular processes. Whether TRIM17 induces or inhibits the ubiquitination of the proteins with which it interacts, its cellular functions, which we review below, depend directly on the nature of its substrates and partners (Figure 8).

### 4.1. Regulation of Transcription Factors

TRIM17 has an impact on transcriptional regulation by modulating the ubiquitination and degradation of transcription factors such as NFATc3 and ZSCAN21 but also by directly inhibiting the activity of NFATc3 and NFATc4 [23,30,65]. 

#### 4.1.1. NFATc3 and NFATc4

The NFAT (Nuclear Factor of Activated T cells) family comprises four calcium/calcineurin-dependent transcription factors that are encoded by four closely related genes [80,81]. These NFAT members are normally found in the cytoplasm in a hyperphosphorylated and inactive state. Upon an increase in intracellular calcium, they are dephosphorylated by the calcium/calmodulin-dependent protein phosphatase calcineurin which triggers their nuclear import and activation [80,81,82]. Once inside the nucleus, NFATs cooperate with multiple transcriptional partners, including activator protein 1 (AP-1), to regulate gene expression [83].

TRIM17 provides an additional level of regulation to these mechanisms by binding to NFATs and preventing their nuclear translocation [23]. Indeed, we found that Trim17 interacts with both NFATc3 and NFATc4 in neuronal cells where they represent the predominant members of the NFAT family. Interestingly, Trim17 appeared to preferentially bind SUMOylated forms of NFATc3, as the coimmunoprecipitation of the two proteins was impaired either by mutations of the three SUMOylation consensus sites of NFATc3 or by alteration of the SUMO interacting motifs (SIMs) of Trim17. In contrast, the interaction between Trim17 and NFATc4, that is shorter and comprises only one SUMOylation site, is SUMO-independent [23]. More importantly, immunofluorescence analyses showed that the nuclear translocation of NFATc3 and NFATc4 triggered by a calcium ionophore, or by neuronal depolarization, was reduced by two-fold following overexpression of Trim17. This was associated with a similar reduction in the activity of the two transcription factors, as estimated by NFATc3 and NFATc4-mediated luciferase expressions and by measuring the mRNA levels of their target gene BDNF [23]. Interestingly, Trim17 could not inhibit NFATc3 nuclear translocation when mutated on its RING domain, suggesting that its E3 ubiquitin-ligase activity may be involved in this effect, although another function of the RING domain cannot be excluded. Moreover, the inhibitory effect of Trim17 on NFATc3 nuclear translocation was abrogated by the mutation of the SIMs of Trim17 and of the SUMOylation sites of NFATc3. As these mutations impair the interaction between the two proteins, these results strongly suggest that Trim17 inhibits NFATc3 by directly interacting with it and thereby preventing its nuclear translocation [23].

Trim17 also acts on the transcriptional activity of NFATc3 by regulating its stability. Indeed, we recently showed that Trim17 inhibits the ubiquitination and proteasomal degradation of NFATc3 mediated by Trim39 [65]. As Trim39 inhibits NFATc3 activity by reducing its protein level [65], Trim17 should favor the transcriptional activity of NFATc3 by relieving this inhibition. Taken together, these data indicate that Trim17 acts on NFATc3 through antagonistic mechanisms. On one hand Trim17 prevents the nuclear translocation of NFATc3, which reduces its transcriptional activity [23], and on the other hand Trim17 should increase the protein level and activity of NFATc3 by inhibiting Trim39 [65]. As Trim17 is itself a target gene of NFATc3 [23], the effects of Trim17 on the protein level and activity of NFATc3 should also influence its own expression, creating both a negative and positive feedback loop.

#### 4.1.2. ZSCAN21

ZSCAN21 (also known as Zipro1/RU49/ZNF38) is a transcription factor regulating the *SNCA* gene which encodes α-synuclein, an abundant presynaptic protein whose deregulation is involved in Parkinson’s disease [30,84,85,86]. Initially identified as a marker for the granule neuron lineage in the CNS [87], ZSCAN21 seems to be expressed throughout the brain, in humans and mice, at different levels, depending on the structure [86,88]. 

We identified ZSCAN21 as a putative binding partner of TRIM17 in a Y2H screen and we have confirmed that the two proteins interact with each other in co-immunoprecipitation and PLA experiments [30]. However, overexpression of TRIM17 does not induce the ubiquitination of ZSCAN21 in cells, indicating that TRIM17 is not an E3 ubiquitin-ligase for ZSCAN21. In contrast, we found that TRIM17 inhibits the ubiquitination and degradation of ZSCAN21 mediated by TRIM41, thereby increasing the stability and the protein level of ZSCAN21 [30]. As a consequence, overexpression of TRIM17 increased the expression of α-synuclein in neuronal cells whereas silencing of TRIM17 reduced it [30]. 

Therefore, by regulating the activity or the protein level of transcription factors, TRIM17 modulates the expression of proteins that play important roles in cell physiology. It is particularly striking for α-synuclein, but it is also certainly important for the target genes of NFATc3 that are involved in many cellular processes.

### 4.2. Apoptosis

Apoptosis is a form of programmed cell death that is evolutionarily conserved. It plays a crucial role in morphogenesis and tissue homeostasis. Importantly, apoptosis is essential for removing cells that represent a threat for the organism, such as precancer cells, infected cells or autoreactive lymphocytes. As a consequence, dysregulations of apoptosis lead to many human disorders. For example, a default of apoptosis is involved in auto-immune and infectious diseases and is a sine qua non condition for tumor progression. In contrast, an excess of apoptosis is evident in AIDS and neurodegenerative diseases [89].

The first cellular function to be attributed to Trim17 was the regulation of apoptosis in neurons [16]. Indeed, we have shown that Trim17 expression is both necessary and sufficient for the initiation of neuronal apoptosis through the intrinsic pathway. Initial observations showed that transfection of primary CGNs with Trim17 leads to cytochrome c release from mitochondria, activation of caspase 3 and cell death. This effect is abolished in neurons deficient in Bax, thereby demonstrating that neuronal death induced by Trim17 results from the specific activation of the intrinsic pathway of apoptosis [16]. Importantly, apoptosis induction by Trim17 depended on its RING domain, suggesting that its E3 ubiquitin ligase activity is required for this function. Moreover, inactive mutants of Trim17, in which the RING domain was deleted or disrupted by a point mutation and that generally exert a dominant-negative effect, protected both CGNs and SCG neurons from apoptosis. These data, that suggest that the activity of Trim17 is necessary for the initiation of neuronal apoptosis, were confirmed by loss-of-function experiments. Indeed, silencing of Trim17 using specific shRNA or siRNA resulted in a strong protection of CGN and SCG neurons from survival factor withdrawal-induced apoptosis [16].

More recently, several CRISPR/Cas9 screens have confirmed the proapoptotic role of TRIM17 and extended it to other cell types. For example, KO of TRIM17 has been shown to confer apoptosis resistance induced by endoplasmic reticulum stress in human fibroblasts [90] and to natural killer cells in chronic myeloid leukemia cells [91]. The mechanisms by which TRIM17 promotes apoptosis are still poorly understood and certainly depend on the cell types and the expression of possible substrates or binding partners. However, a few studies have started to elucidate some of these mechanisms, notably in neurons.

#### 4.2.1. MCL1

The first mode of action that was identified to explain the proapoptotic effect of TRIM17 was the ubiquitination and degradation of the antiapoptotic protein MCL1. Indeed, MCL1 is the only substrate of the E3 ubiquitin-ligase activity of TRIM17 that has been formally identified so far [72]. MCL1 is an important member of the Bcl-2 family. This family comprises both antiapoptotic (BCL2, BCL-xL, MCL1, BCL2A1, etc.) and proapoptotic (BAX, BAK, BH3-only proteins, etc.) members that play a pivotal role in the regulation of the intrinsic pathway of apoptosis by negatively or positively controlling the release of cytochrome c from mitochondria and thereby the activation of caspases [92]. MCL1 is essential for the survival of multiple cell lineages and is highly amplified in human cancers [93]. Under physiological conditions, MCL1 expression is tightly regulated at multiple levels, involving transcriptional, post-transcriptional, translational and post-translational processes [23,94]. MCL1 is characterized by a short half-life. Its ubiquitination, that targets it for proteasomal degradation, allows for rapid removal of MCL1 and initiation of cell death, in response to various cellular events [23]. Several E3 ubiquitin-ligases have been identified for MCL1 [94,95], including TRIM17. Indeed, we have shown TRIM17 to be an E3 ubiquitin-ligase for MCL1 in neurons [72]. Indeed, TRIM17 binds to MCL1 in coimmunoprecipitation experiments. Interestingly, this interaction depends on the GSK3-mediated phosphorylation of mouse Mcl1 on Ser140 and Thr144, two residues that favor the ubiquitination and degradation of MCL1 when phosphorylated [72]. In addition, TRIM17 is able to ubiquitinate MCL1 in vitro, in a completely acellular medium, indicating that MCL1 is a direct substrate of Trim17. Furthermore, overexpression of TRIM17 decreases the protein level of MCL1, this effect being reduced by the mutation of the critical phosphorylation sites of MCL1. In contrast, silencing of TRIM17 expression both reduces the ubiquitination level of MCL1 and increases its half-life [72]. Taken together, these data indicate that TRIM17 is a physiological E3 ubiquitin-ligase of MCL1 in neurons. The ubiquitin-mediated elimination of MCL1 may therefore underly, at least in part, the proapoptotic effect of Trim17. Consistently, the mutation of the phosphorylation sites of MCL1 that are involved in its interaction with Trim17 increased MCL1 antiapoptotic effect in neurons by improving its stability [72].

Interestingly, an independent study aiming at elucidating the molecular basis of immune resistance in cancer cells showed that NANOG-mediated/HDAC1-driven epigenetic silencing of TRIM17 resulted in stabilization of MCL1 and subsequent resistance to apoptosis in immune-edited tumor cells [22]. Therefore, the role of TRIM17 in MCL1 degradation and apoptosis regulation is not restricted to neurons and may also be important in other physiological functions such as immune surveillance of tumorigenesis.

#### 4.2.2. BCL2A1

TRIM17 also contributes to apoptosis regulation by modulating the stability of another antiapoptotic protein of the BCL2 family that is the closest phylogenetic homolog of MCL1: BCL2A1. Like MCL1, BCL2A1 has a short half-life due to its constitutive degradation by the ubiquitin-proteasome system. We recently identified TRIM28 as an E3 ubiquitin-ligase of BCL2A1 and have shown that GSK3 is involved in the phosphorylation-mediated inhibition of BCL2A1 degradation [69]. Interestingly, TRIM17 binds to BCL2A1 but does not induce its ubiquitination. On the contrary, TRIM17 stabilizes BCL2A1 by preventing TRIM28 from binding and ubiquitinating BCL2A1. This inhibitory effect results in the stabilization of BCL2A1 and increased resistance of melanoma cells towards apoptosis [69]. In this specific case, TRIM17 therefore exerts an antiapoptotic effect. It should be noted that BCL2A1 and MCL1 are regulated by common factors, GSK3 and TRIM17, but with opposite outcome. This might be related to the presence in BCL2A1 of an α-helix coming from the duplication of a sequence attributed to the proapoptotic protein HCCS-1 [96]. It is conceivable that this duplication event brought GSK3 and TRIM17-mediated stabilization to BCL2A1 while it originally stabilized a proapoptotic protein. Whatever the reason for this paradox, TRIM17 may have a proapoptotic effect in neurons where MCL1 expression is crucial for survival and have an antiapoptotic effect in cell types expressing BCL2A1, the two antiapoptotic proteins generally showing different patterns of tissue expression [69,72].

#### 4.2.3. NFATc3 and NFATc4

In addition to its effect on BCL2 family proteins, TRIM17 can regulate apoptosis by acting on NFAT transcription factors. Indeed, NFATs are involved in a wide range of cellular and physiological processes including apoptosis [81,97,98]. Notably, NFATc3 and NFATc4 have been reported to play an important role in the control of the survival/death fate of neurons [99,100,101,102]. Indeed, data from gain of function and loss of function experiments [23,99,100] or in NFATc4-deficient mice [101], suggest that NFATc4 promotes survival in different types of neurons, in particular by inducing the transcription of survival factors [23,100]. As mentioned above, TRIM17 directly inhibits NFATc4 activity by preventing its nuclear translocation. Therefore, this inhibition may partially mediate the proapoptotic effect of TRIM17 in neurons [23]. The role of NFATc3 in neuronal death is less documented. However, two studies suggest that NFATc3 has a proapoptotic effect [23,100]. It is nevertheless difficult to anticipate the exact impact TRIM17 can have on apoptosis through its antagonistic functional interactions with NFATc3 (see above). Indeed, TRIM17 can both directly inhibit NFATc3 by blocking its nuclear translocation [23] and increase its protein level by inhibiting its TRIM39-induced degradation [65]. Moreover, TRIM17 expression is itself induced by NFATc3 [23], therefore creating complex negative and positive feedback loops that should eventually result in fine tuning of neuronal apoptosis.

Therefore, although initially identified as a proapoptotic protein, the role of TRIM17 in cell death regulation is certainly much more complex, TRIM17 being able to both induce and inhibit apoptosis. This fascinating duality of action is expected to depend on cell type and physiological context that dictate the expression and the role of its possible substrates and partners.

### 4.3. Autophagy

Autophagy is a homeostatic mechanism by which eukaryotic cells remove toxic aggregates, invading pathogens, damaged and excess organelles, or recycle cytosolic components to meet metabolic requirements [103]. During autophagy, cytoplasmic materials are enclosed in double-membrane vesicles called autophagosomes and are further delivered to lysosomes for degradation [103,104]. In the cytosol, the UPS and autophagy lysosome (AL) pathways act simultaneously. They share components of their molecular machineries and influence each other’s activities [13,105,106]. Notably, the autophagy receptor sequestosome-1 (SQSTM1)/p62 is the main molecule that regulates the cross-talk between the two systems [13,107]. 

TRIM proteins have been shown to play a major role in the regulation of autophagy, both in physiological and pathological conditions, by acting as autophagy regulators and as autophagy receptors that bind targets to be degraded [108,109,110]. They provide a potential link between the specificity and the regulation of autophagy [111]. Despite its interaction with a subset of protein complexes involved in autophagy induction (mAtg8s, p62 and ULK1-Beclin-1), TRIM17 was first identified as a negative regulator of basal and mTOR inhibition-induced autophagy, because its knockdown dramatically increased the number of autophagosomes in cells [109]. In fact, this effect on autophagosomes is due to the ability of TRIM17 to recruit autophagic factors in a localized and limited area of the cell [109]. In addition to this role in bulk autophagy, TRIM17 inhibits selective autophagic degradation of a subset of targets, such as p62, IFT20, TRIM5α or TRIM22, while promoting the degradation of at least one other target, the midbody, the transient structure that connects two daughter cells at the end of cytokinesis [68]. Interestingly, the function of TRIM17 in selective autophagy regulation involves its interaction with MCL1. Indeed, MCL1 exerts an antiautophagy activity in addition to its antiapoptotic role by binding and inactivating Beclin 1 [112], a key inducer of autophagy. TRIM17 inhibits selective autophagy by stabilizing the MCL1/Beclin 1 complex. In contrast, when TRIM17 exerts a proautophagy function, it releases MCL1 from the complex that it forms with Beclin 1, thereby disinhibiting autophagy at defined sites. Therefore, the presence or absence of MCL1 determines whether TRIM17 complexes inhibit or promote selective autophagy [68].

Moreover, TRIM17 has been shown to interact with both Galectin-3 and Galectin-8 which participate in the autophagic response to endomembrane perforation caused by lysosomal damaging agents and by bacteria, suggesting a possible role of TRIM17 in this process [113].

### 4.4. Cell Proliferation and Mitosis

Many TRIM proteins have been associated with cell cycle phase transitions and mitotic progression [114]. Among the different phases of the cell cycle, mitosis is a delicate event in which the segregation of chromosomes into two daughter cells takes place. It is very important that each daughter cell receives an exact copy of the genetic material, and defects in chromosome segregation have been associated with tumorigenesis [115]. During mitosis, centrosome duplication and subsequent centrosome separation ensure the formation of the mitotic spindle, consisting of microtubules [116]. To ensure the precise distribution of DNA during mitosis, the kinetochore macromolecular complex assembles on the centromere of each chromosome [117] and its interface with microtubules enables chromatid segregation [118]. 

For most TRIM proteins, silencing generally increases the percentage of cells in G0/G1 and reduces cells in the S or G2-M phase. Two independent studies suggest that TRIM17 plays a role during mitosis progression [68,73]. Indeed, in a yeast two-hybrid screening assay using TRIM17 as bait, Endo et al. identified ZW10 interacting protein (ZWINT), a known component of the kinetochore complex required for the mitotic spindle checkpoint, as a putative TRIM17 partner [73]. TRIM17 overexpression induced proteasomal degradation of ZWINT and the coiled-coil domain of TRIM17 was found to be required for the TRIM17/ZWINT interaction. These results suggest that ZWINT could be a substrate of TRIM17 although the precise mechanism by which TRIM17 induces the degradation of ZWINT was not examined. Moreover, overexpression of TRIM17 decreased cell proliferation in MCF7 cells in the same manner as ZWINT knockdown, suggesting that TRIM17 may interfere with mitosis by inducing the degradation of ZWINT [73]. In another study mentioned above and addressing the role of TRIM17 in autophagy, the DNA content of TRIM17 knockdown cells was assessed. A higher percentage of cells were in S and/or G2 phase after knockdown of TRIM17 compared to cells transfected with a control siRNA, suggesting that TRIM17 promotes cell division [68]. Accordingly, we observed that stable TRIM17 knockdown, using shRNA-expressing lentiviruses, resulted in decreased cell proliferation and aberrant mitosis with signs of endoreplication and cytokinesis defects (unpublished data). Taken together, these results suggest that TRIM17 can exert opposite effects on cell cycle regulation which might be substrate and context dependent. 

Current data therefore indicate that TRIM17 is a pleiotropic protein that plays important roles in the regulation of cellular processes as crucial as apoptosis, autophagy and cell proliferation. The cellular functions of TRIM17 result from both indirect and direct actions on key players in these processes. By regulating the activity of transcription factors, either by directly binding them or by modifying their stability, TRIM17 modulates the expression of important genes such as *SNCA*. TRIM17 can also directly target key proteins involved in cell division, autophagy of cell death and modulate their protein level, either by inducing or inhibiting their ubiquitination. Much remains to be discovered about the modes of action of TRIM17, but there is no doubt that their elucidation will lead to a better understanding of the mechanisms governing cell life and death.

### 4.5. TRIM17 Knockout Mice

Using CRISPR/Cas9 genome editing, Lu et al. have generated knockout mice for 30 testis-enriched genes, including TRIM17, to study their roles in spermatogenesis, sperm function and in male reproduction [18]. To knockout all variants of TRIM17, they designed sgRNAs to target the first coding exon (exon 2) and the 3′ UTR region. They observed no abnormal development or behavior in the generated homozygous mouse lines and no significant differences in testis weight or size were observed between the KO and wild-type mice. As TRIM17 is highly expressed in testis, histological examination of testes and epididymides in wild-type and knockout mice was performed. No abnormalities were observed in terms of composition, quantity, or morphology of spermatogenic cells in the seminiferous tubules. Equally, spermatozoa in the caput and cauda epididymides were normal and in the motility of Trim17-null spermatozoa was comparable to that of wild-type [18]. 

TRIM17 (Trim17^tm1e(EUCOMM)Wtsi^) knockout mice were also created by The International Mouse Phenotyping Consortium (IMPC). However, during the engineering of the embryonic stem cells used to generate these mice, the third loxP site located between exon 3 and exon 4 of TRIM17 was lost due to recombination events. This prevents the removal of exons 2 and 3 necessary for the conditional KO of TRIM17. Apparently, the LacZ-tagged alleles could report endogenous gene expression and are highly likely to be null mutations [119]. The phenotype observed for these mice is limited to increased circulating magnesium, increased thyroxine and alkaline phosphatase levels together with an abnormal pelvic girdle bone morphology (www.mousephenotype.org, 1 March 2021). It is currently difficult to relate this phenotype to the known cellular functions of TRIM17. In addition, the genotype of these mice has not been specifically described and no information on the actual expression of TRIM17 is available. Therefore, new data confirming the efficacy of TRIM17 KO are needed to allow further study of these mice.

Although previous studies in cells suggest that TRIM17 should have a crucial role during development by controlling cell proliferation and apoptosis, particularly in brain, these two studies surprisingly indicate that TRIM17 deletion does not induce major defects in mouse development. This raises the question of a possible redundancy that would compensate for the absence of TRIM17 during development. However, despite the high degree of structural similarity, the overall average sequence identity between TRIMs with PRY-SPRY domain is not greater than 35–45% even if some residues are highly conserved [120]. The protein that substitutes for TRIM17 therefore remains to be determined. Generation of conditional KO mice that allow deletion of *Trim17* in specific organs or at different developmental stages may help answer this question. Concerning the lack of phenotype in adult TRIM17 KO mice, it is important to keep in mind that TRIM17 is expressed at very low levels, in normal conditions, in most tissues. The phenotype of adult KO mice may therefore be revealed only in stress conditions, when TRIM17 expression is highly induced, notably to fulfill its proapoptotic function. Therefore, it would be really exciting to subject these transgenic mice to different toxins or stresses to determine whether the absence of TRIM17 confers a resistance in these conditions.

## 5. TRIM17 and Diseases

### 5.1. Parkinson’s Disease

Parkinson’s disease (PD) is the second most common neurodegenerative disorder after Alzheimer’s disease and it is an important cause of chronic disability [121]. It results primarily from the selective degeneration of dopaminergic (DA) neurons in the substantia nigra pars compacta (SNpc). Although most cases of PD are sporadic with a variable contribution of both environmental and genetic factors, about 10% of cases are associated with mutations in genes with autosomal dominant or recessive inheritance. *SNCA* which encodes the presynaptic protein α-synuclein, is arguably the most important gene linked to PD [122,123] and accumulating data indicate that increased expression of wild-type α-synuclein plays a crucial role in PD neurodegeneration [124,125].

TRIM17 is potentially implicated in the pathogenesis of Parkinson’s disease in several ways. First, by showing that ZSCAN21 is a transcription factor for *SNCA* and that its stability is antagonistically regulated by TRIM41 and TRIM17 (see above), we have implicated TRIM17 in a new pathway regulating α-synuclein expression [30]. Indeed, overexpression of TRIM17 stimulates, whereas silencing of TRIM17 reduces α-synuclein expression in a neuronal cell line [30]. Moreover, in the MPTP-based mouse model of PD, an increase in TRIM17 expression is concomitant with the already reported increase in α-synuclein expression. These data therefore suggest that TRIM17 induction may be involved in PD pathogenesis related to environmental factors, such as the neurotoxin MPTP or chemically related pesticide or herbicide molecules, by increasing α-synuclein expression [30]. Consistent with a possible role of TRIM17 in α-synuclein expression and its deregulation, analysis of the GTEX cohort with Xena browser revealed a large correlation between high expression of *TRIM17* and high expression of *SNCA* genes [126].

Second, a few genetic variations in the *TRIM17* gene were found in PD patients. Indeed, we identified two rare variants of *TRIM17* in a cohort of 200 patients with autosomal dominant PD: TRIM17p.R78W and TRIM17p.T407N [30]. Interestingly, the amino acid substitution T407N is located in the PRY-SPRY domain of TRIM17 which is thought to be involved in the interaction between TRIM proteins and their partners [39], as it is the case for TRIM41 and ZSCAN21 [30]. This genetic variation may therefore impair the interaction of TRIM17 with its partners and affect the TRIM17/TRIM41/ZSCAN21 pathway. The R78W substitution is located in a region linking the RING and the B-Box domain, called RBL linker [120]. This region is highly conserved throughout species in TRIM17 and is also present in other TRIM proteins, suggesting an important functional role. Interestingly, the RBL motif has been shown to be critical for the folding of the entire RING-B-box region of TRIM21 [120]. Consistently, in a whole-exome sequencing study aiming at identifying genetic variants contributing to disease risk in familial PD, two nonsynonymous variants of *TRIM17* (TRIM17p.Q71E and TRIM17p.L194P) were identified in a cohort of 93 individuals from 32 families with PD [127]. It is important to note that the Q71E substitution is located in the RBL linker and L194P in the coiled-coil domain of TRIM17. As discussed above, these two regions are important for TRIM functions. Frequency analyses of these four TRIM17 variants showed that they are all very rare in the general population and three of them are predicted to be damaging or possibly damaging for the protein function by SIFT or Mutation taster algorithms [127]. However, despite the bioinformatic filters applied in the study from Farlow et al. to predict damaging alleles, family members carrying the TRIM17 variants also exhibited variants in eight other genes. Further studies are thus needed to elucidate the functional consequences of these TRIM17 variations and to determine whether they have any causal link with PD.

Finally, as mentioned previously, TRIM17 directly inhibits and prevents the degradation of another transcription factor, NFATc3 [23,65]. Two independent studies have implicated NFATc3 in α-synuclein-induced degeneration of midbrain dopaminergic neurons in PD [128,129]. However, further investigations are needed to determine whether TRIM17 may participate in the toxicity of α-synuclein by acting on NFATc3.

### 5.2. Autism

Genetics is a major contributor to autism spectrum disorders (ASD), de novo mutations representing an important source of causality [130]. Three high throughput DNA sequencing studies implicate TRIM17 de novo mutations in ASD [131,132,133]. Notably, an insertion in the *TRIM17* gene located just before the sequence coding for the PRY-SPRY domain was linked to autism in two studies [132,133]. This insertion leads to the expression of a truncated form of TRIM17 deleted of its PRY-SPRY domain. In a recent large-scale sequencing study from a cohort of 35584 samples, including 11986 samples with ASD, 102 genes implicated in risk for ASD were identified [131]. In this study, a missense variation located in the coiled-coil domain of TRIM17 (TRIM17p.R159T) was associated with ASD [131]. These two genetic variations in *TRIM17* affect two domains that are important for TRIM activity: the coiled-coil domain allowing TRIM dimerization and the PRY-SPRY domain usually involved in the interaction with substrates or partners. This suggests that TRIM17 activity could be altered by these variations. Further investigations are needed to determine if *TRIM17* is actually linked to ASD and by which mechanism. However, it is interesting to note that genetic variations in two partners of TRIM17, ZSCAN21 [134] and TRIM41 [131], were also linked to ASD in independent studies. Notably, the variation that was identified in TRIM41 affects its PRY-SPRY domain [131]. In addition, *SNCA* gene deletions and partial duplication have been found in ASD patients [135,136]. As accumulating arguments suggest that α-synuclein may be involved in ASD pathogenesis [136], it is tempting to speculate that TRIM17 may also play a role in ASD by altering α-synuclein expression through its action on TRIM41 and ZSCAN21.

### 5.3. Cancer and Chemoresistance

Tumorigenesis is a multistep process. Each step reflects genetic alterations that confer one or another type of growth advantage, leading to the progressive transformation of normal cells into highly malignant derivatives. Among the essential cellular process whose alteration participates in tumorigenesis [137], TRIM17 is involved in apoptosis and cell proliferation. In particular, TRIM17 regulates the protein levels of two antiapoptotic proteins of the Bcl-2 family, MCL1 and BCL2A1, that both act as oncogenes. Indeed, MCL1 is one of the most highly amplified genes across a variety of solid and hematological human malignancies [93]. In many cancers, MCL1 is essential for cancer cells to overcome oncogenic stress-induced apoptosis [138] and for chemotherapeutic resistance and relapse [139,140]. Similarly, BCL2A1 is highly upregulated in several hematopoietic malignancies and melanoma, in which it contributes to chemoresistance [141,142]. Downregulation of BCL2A1 in these malignancies restores sensitivity to chemotherapeutics [143,144]. TRIM17 is also involved in mitosis by inducing the degradation of key components of the mitotic spindle machinery, midbodies and the kinetochore protein ZWINT, via autophagy and UPS, respectively [68,73]. Notably, ZWINT is upregulated in many human cancers and is associated with poor clinical prognosis and early recurrence [145,146]. Conversely, ZWINT knockdown effectively inhibits proliferation of glioblastoma cells in vitro and suppresses glioblastoma growth in vivo [146]. Therefore, it would not be surprising that dysregulation of TRIM17 participates in tumorigenesis and chemoresistance by altering apoptosis and mitosis regulation. Some data support this hypothesis.

TRIM17 expression has been shown to be modified in tumors [147,148]. For example, TRIM17 expression was found to be significantly increased in osteosarcoma tissues compared with adjacent nontumorous tissues [147] and TRIM17 is the most frequently amplified of the TRIM genes in breast invasive carcinomas [149]. Analysis of TCGA (The Cancer Genome Atlas program) data based on the correlation between mRNA expression levels (FPKM values) of TRIM17 and patient survival in breast cancers showed that higher expression of TRIM17 is associated with a better survival probability and that TRIM17 expression could be used as a favorable prognostic indicator in breast cancers [150] (www.proteinatlas.org). However, additional investigations are needed to understand whether these increases are a cause or a consequence of the transformed phenotype.

Several genome-wide KO screens by CRISPR/Cas9 have identified TRIM17 as one of the genes whose inactivation results in drugs resistance in tumor cell lines. For example, KO of TRIM17 confers resistance to the BCL2 inhibitor Venetoclax in acute myeloid leukemia cells [151], to the bacterial toxin T3SS2 in a Colonic Adenocarcinoma Cell line [152] and to the ferroptosis-inducing drug ML210 in clear-cell carcinoma cell lines [153]. However, the mechanisms by which TRIM17 confers drug sensitivity in these tumor cells has not been addressed. In the case of immune resistance conferred by the NANOG-HDAC1 axis (see above), the effect of the epigenetic silencing of TRIM17 was clearly attributed to MCL1 stabilization [22]. Although most studies point to a drug sensitization effect, TRIM17 may also play a role in chemoresistance in some tumors. Indeed, we found that TRIM17 expression is induced in melanoma cells following treatment with the BRAF inhibitor PLX4720. More interestingly, KO of TRIM17 restored sensitivity to PLX4720 in resistant melanoma cells, by relieving the inhibition of TRIM28-mediated ubiquitination of BCL2A1 and thereby by reducing BCL2A1 level [69]. Therefore, TRIM17 may be involved in the chemoresistance of cancer cells that exhibit a survival dependency on BCL2A1 whereas it would increase the sensitivity of tumors depending on MCL1, due to its opposite effects on these two antiapoptotic proteins (see above).

Additional investigations are required to determine the relative importance of TRIM17 in tumorigenesis and drug resistance. However, existing data clearly indicate that the role TRIM17 can have in cancer depends on tumor types, the mechanisms underlying transformation and resistance, and the nature of the proteins that it can target in these pathways. 

### 5.4. Other Pathologies

Several variants of the *TRIM17* gene have been identified that are potentially associated with different additional pathologies. For example, a candidate-gene testing for orphan limb-girdle muscular dystrophies (LGMD) identified a single mutation in exon 4 of the *TRIM17* gene in patients [154]. However, the association of this mutation with LGMD phenotype remains to be investigated. In a recent study, next-generation sequencing of 130 children with hypospadias of unknown etiology identified the variant TRIM17p.R37X in one hypospadias patient. However, this patient also carried the ARp.Q58L variant, making it difficult to establish a link between the TRIM17 variant and the phenotype [155]. In addition, high throughput screenings have linked mutations in the *TRIM17* gene with various disorders. For example, the variants TRIM17p.V132L and TRIM17p.G326V have been found to be associated with multiple myeloma [156] and recessive cognitive disorders [157], respectively. 

Although many studies are necessary to establish the pathological role of TRIM17 in neurological diseases, cancer or other disorders, its implication in cellular processes that are dysregulated in many pathologies, such as cell death and proliferation, makes it an obvious candidate. Importantly, TRIM17 is a particularly interesting therapeutic target. Indeed, as it is specifically induced in pathological conditions, its inhibition should have less detrimental side effects. Moreover, since its cellular functions are generally related to its interaction with defined partners, interfering with these specific interactions should inhibit its pathological effects without altering its other functions.

## 6. Conclusions

TRIM17 is a pleiotropic protein at the crossroad of many crucial cellular functions such as apoptosis, autophagy and cell cycle. Fascinatingly, it can play an opposite role in these functions, depending on the partners it interacts with. Its expression is generally very low, and it is essentially only detectable in brain and testes in adulthood. However, it can be highly induced by cellular stress, anticancer drugs or exposure to environmental toxins in many tissues, making it a decisive trigger signal in various pathophysiological conditions. Although TRIM17 possesses E3 ubiquitin ligase activity to drive the degradation of substrates such as MCL1 and ZWINT, it also functions as a stabilizer of NFATc3, BCL2A1 and ZSCAN21 by inhibiting TRIM28, TRIM39 and TRIM41, respectively. This duality of molecular function may be explained by the ability of TRIM proteins to associate with several E2 enzymes and different TRIM partners, including their own isoforms. The protein complexes assembled by TRIM17 could either promote or inhibit the binding of substrates or E2 enzymes, eventually leading to the stabilization or degradation of the target. A major challenge for a better understanding of the cellular and pathophysiological roles of TRIM17 will be to elucidate the other proteins whose stability it regulates, as well as the composition of TRIM17 complexes and the cellular context in which they form. This is of particular importance, as the specific interaction with its substrates or its TRIM partners may be targeted in pathological conditions in which TRIM17 is induced, for the treatment of human diseases as diverse as PD and cancers.

## Figures and Tables

**Figure 2 cells-10-01235-f002:**
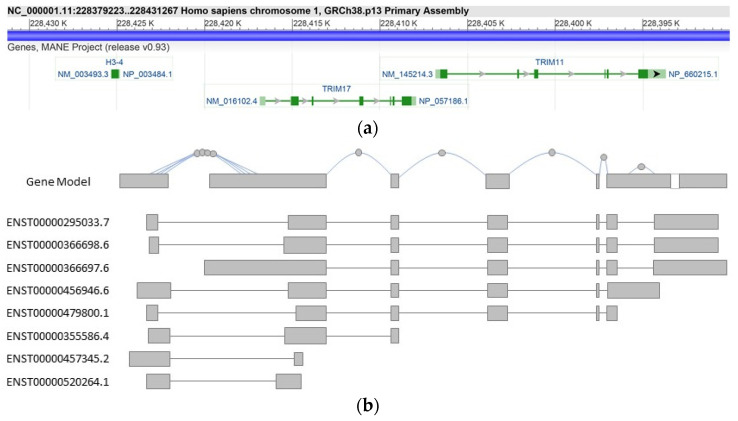
(**a**) Screenshot from NCBI’s Genome Data Viewer. The data track MANE Project (release v0.93) shows TRIM17 and TRIM11 genes’ localization and transcripts (NCBI’s Refseq names NM_016102.4 and NM_145214.3, respectively) on GRCh38.p13 primary assembly; (**b**) gene model (ENSG00000162931.11) and TRIM17 mRNA isoforms (from gtexportal.org, 21 February 2021, Source: HGNC Symbol; Acc: HGNC:13430).

**Figure 3 cells-10-01235-f003:**
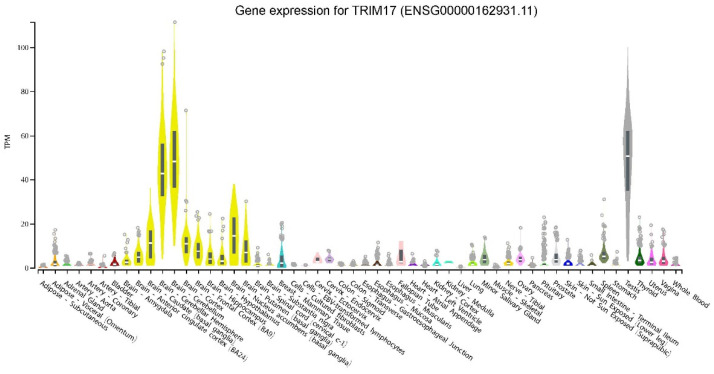
Cell type expression profile of human *TRIM17* gene (from https://gtexportal.org/home/, 21 February 2021). Data source: GTEx Analysis Release V8 (dbGaP Accession phs000424.v8.p2); TPM: transcripts per million (for additional information, see https://academic.oup.com/bioinformatics/article/26/4/493/243395).

**Figure 4 cells-10-01235-f004:**
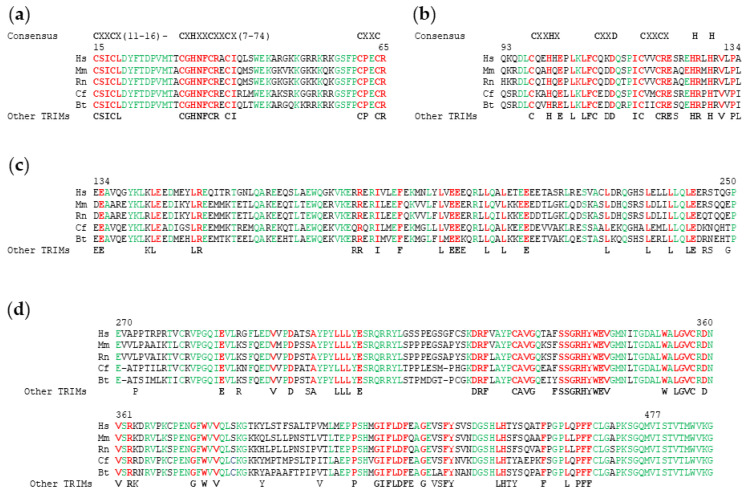
Species conservation of TRIM17 RING (**a**), Bbox2 (**b**), coiled-coil (**c**) and PRY-SPRY (**d**) domain sequences and comparison with other TRIMs and known consensus sequences. In bold, TRIM17 conserved residues in consensus sequence or shared by other TRIMs. In red, conserved residues through species. In green, residues highly conserved through species possibly involved in specificity of TRIM17.

**Figure 5 cells-10-01235-f005:**
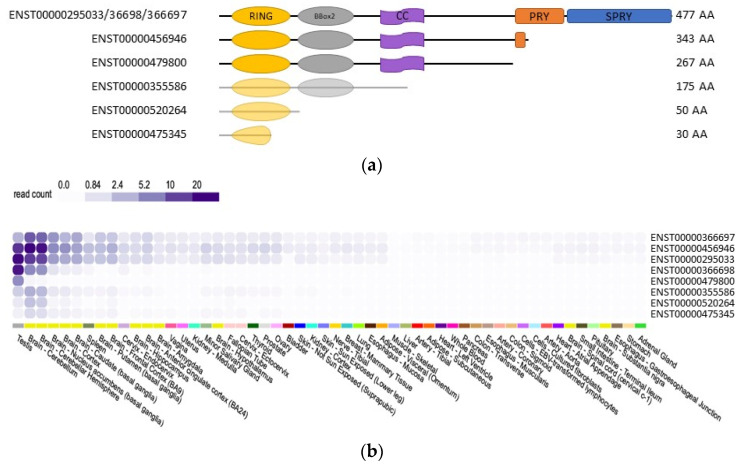
(**a**) Schematic representation of the expected domain architecture of TRIM17 isoforms and the corresponding protein coding lengths (in amino acids) (from ensembl.org, 21 February 2021); (**b**) expression profile of TRIM17 isoforms [Source:HGNC Symbol;Acc:HGNC:13430].

**Figure 6 cells-10-01235-f006:**
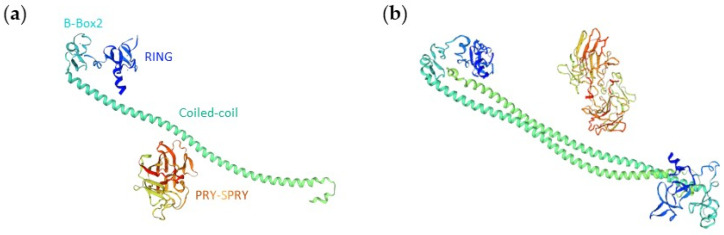
(**a**) Predicted monomeric 3D structure of tripartite motif and PRY-SPRY domains of human TRIM17 (Q9Y577) based on TRIM28 (6QAJ.1A) and PYRIN (2WL1.1A), respectively (from Swiss Model Repository database); (**b**) predicted 3D structure of TRIM17 homodimers based on homodimers of tripartite and PRY-SPRY domains of TRIM28 and PYRIN respectively (from Swiss Model Repository database).

**Figure 7 cells-10-01235-f007:**
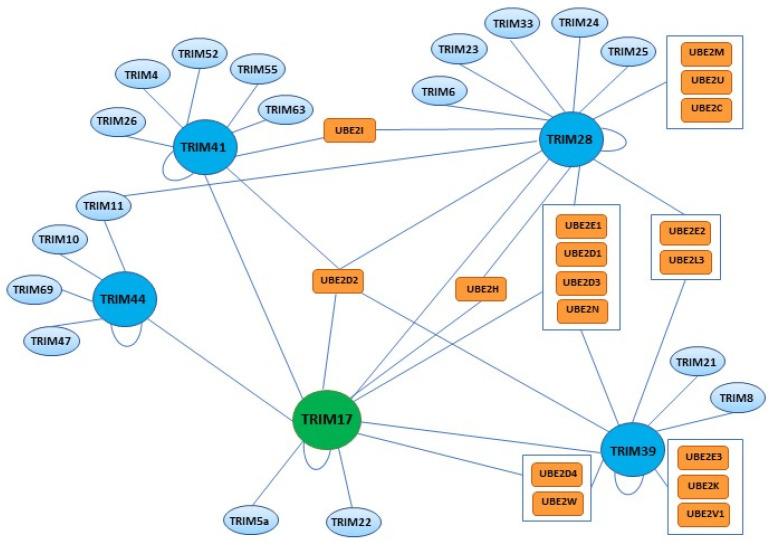
TRIM and E2 partners of TRIM17 from experimental data.

**Figure 8 cells-10-01235-f008:**
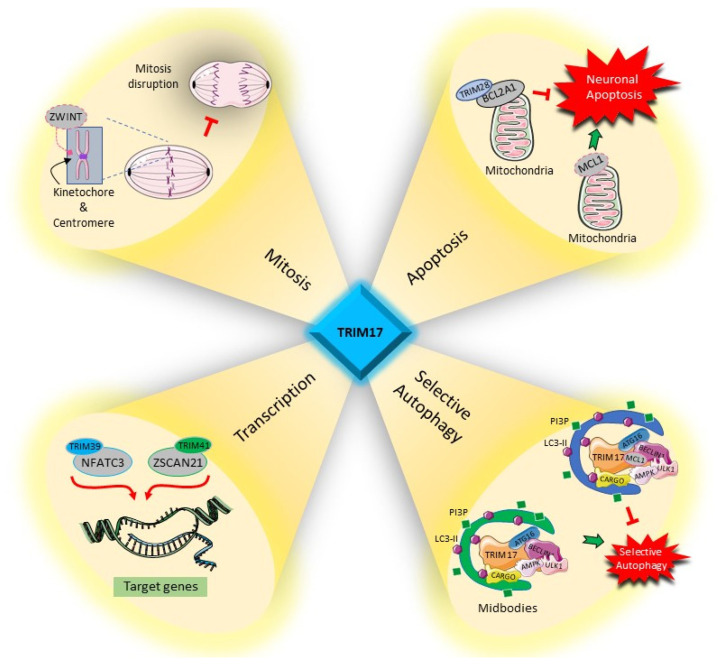
Cellular functions of TRIM17.

## Data Availability

Not applicable.

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
