# Peer review of "To Ubiquitinate or Not to Ubiquitinate: TRIM17 in Cell Life and Death"

_cells, 2021, doi:10.3390/cells10051235_

Round 1
Reviewer 1 Report
Manuscript by Shrivastrava et al entitled “Ubiquitinate or not ubiquitinate: TRIM17 in cell life and death” reviews structure and function of a specific member of the TRIM family, namely the TRIM17. Authors discuss in depth the structure and regulation of the TRIM17, summarize the structure and functions of TRIM17, describe cellular functions of TRIM17 and last, elaborate on possible involvement of TRIM17 in various disorders. The review is well written, albeit few linguistics mistakes need to be fixed, and most informative. At the same time, however, the manuscript by Shrivastrava et al in consideration for publication in Cells reads too factual and is lacking in its section a more personal interpretation of the currently published results. In short, a bit of a personal touch would make the reading significantly more interesting.
Author Response
We really thank the reviewer for his/her comments. We agree that the review was sometimes too factual and needed a « personnal touch ». Therefore, we have added some personal interpretations:
- Page 2, lines 61-62: This weak expression, which makes it difficult to detect with antibodies, may explain why TRIM17 was not studied for a long time.
- Page 2, lines 67-69: Although it is low in most situations, TRIM17 expression can be dramatically induced following different cellular stresses, giving it the role of a sentinel ready to trigger the appropriate cellular response.
- Page 6, lines 217-218: The identification of the transcription factors and regulatory elements that control its expression is all the more crucial.
- Page 13, lines 473-475: However, this duality, whereby TRIM17 can promote or inhibit the ubiquitination and degradation of specific proteins depending on the cellular context, already appears to be crucial for its cellular functions, as discussed below.
- Page 17, we have modified two sentences in lines 668-669.
- Page 18, lines 740-749: we have added a whole paragraph to insist on the particular role of TRIM17 and the interest to study it.
- Page 20, lines 787-789: Generation of conditional KO mice that allow deletion of Trim17 in specific organs or at different developmental stages may help answer this question.
- Page 20, lines 793-795: Therefore, it would be really exciting to subject these transgenic mice to different toxins or stresses to determine whether the absence of TRIM17 confers a resistance in these conditions.
- Page 23, we have added a whole paragraph to emphasize the possibility of targeting TRIM17 in highly specific therapies.
- Page 23, lines 962- 963: Fascinatingly, it can play an opposite role in these functions, depending on the partners it interacts with.
We also asked a native English-speaking colleague to go through our manuscript to identify and correct linguistic and spelling errors.

Reviewer 2 Report
The authors present a comprehensive review on the characteristics of TRIM17, a member of the Tripartite Motif family of E3-ubiquitin ligases. The cover the structure and regulation of the gene encoding TRIM17, and also the protein structure and molecular function of this protein. They also review the cellular functions of TRIM17 at different levels: regulation of the activity of transcription factors, apoptosis, autophagy and cell proliferation. Finally, they explore the possible roles of TRIM17 in diseases such as Parkinson’s, autism and cancer.
The review is sound and very updated to the present knowledge on TRIM17.
I have only minor comments that are related to the quality of the figures.
-In Fig 1, some of the colors of the different domains of the TRIM proteins do not allow to see the description of the domain. I would suggest to either change the colors or put the words in white to distinguish the names of the domains.
-In Fig 8, the red color does not allow to see the words.
Author Response
I have only minor comments that are related to the quality of the figures.
-In Fig 1, some of the colors of the different domains of the TRIM proteins do not allow to see the description of the domain. I would suggest to either change the colors or put the words in white to distinguish the names of the domains.
We agree with the reviewer. We put the color of the writing in white and the font in bold to better distinguish domain names.
-In Fig 8, the red color does not allow to see the words.
As requested by the reviewer, we put the color of the writing in white to better see the words inside red objects.
